# RESET METHOD BASED ON THE THEORY OF MANIFOLD OPTIMIZATION ON REAL MANIFOLDS

## ABSTRACT

In the fields of applied mathematics, statistics, and machine learning, particularly deep learning, manifold optimization assumes a prominent role. By leveraging the intrinsic geometric properties of manifolds, the endeavor to solve constrained optimization problems can be equivalently transformed into the pursuit of unconstrained optimization problems over manifold structures. However, manifold optimization is slow to converge and unstable. To address these issues, an innovative method is introduced, named Reset method that combines the other methods (SGD,Adam and AdamW), aiming to enhance both the improvement of precision and the reduction of convergence loss. Subsequently, our approaches have yielded notable outcomes in terms of the improvement of precision and enhanced stability of the model. The efficacy of our proposed methods is corroborated by the results of deep learning experiments, which provide compelling evidence in support of our initial hypothesis.

## 1 INTRODUCTION

Fast and stable optimization methods have long been pursued by numerous researchers across different generations. Stochastic gradient-based optimization methods, exemplified by stochastic gradient descent (SGD), have achieved notable success in various domains. However, their convergence rate remains relatively slow. In recent years, there has been a proliferation of innovative approaches aimed at accelerating optimization through the application of adaptive learning rates. Notably, Adam Kingma & Ba (2014) and AdamW Loshchilov & Hutter (2017) have gained widespread adoption due to their ability to facilitate rapid convergence.

Over the past few years, research into the manifold optimization has started to gain increasing attention. Several methods for manifold optimization have been proposed by leading scientists, such as trust domain method Absil et al. (2008), adaptively regularized Newton's method Wu et al. (2017) , quasi-Newton-type method Huang et al. (2015), Broyden–Fletcher–Goldfarb–Shanno (BFGS) method Huang et al. (2018) and Stochastic Variance Reduced Gradient (SVRG ) method Zhang et al. (2016), and have established plenty of performance evaluation and analysis methods. In addition, some scholars have provided vast open source software packages, such as the Convex Optimization Simulation System CVX, the First-order Conic Curve Solver Paradigm TFOCS software, etc. There are numerous problems that are large in size themselves, but the data itself may fall on a low-dimensional manifold in a high-dimensional space, and thus can be transformed into an optimization problem on the manifold. Manifold optimization has been the subject of considerable attention and has been employed extensively in a number of fields, such as Douik & Hassibi (2022) and Hu et al. (2020), including computational mathematics, applied mathematics, statistics and machine learning. Of these, particular emphasis has been placed on the fields of deep learning.

However, by leveraging the inherent geometric structure of the manifold, a significant constrained optimization problems, such as Ergen et al. (2022) and Ergen & Pilanci (2021), can be transformed into unconstrained optimization Hu et al. (2020) problems on the manifold itself. Extensive investigations have been conducted to explore the algebraic and topological structure, optimization conditions, and numerical analysis associated with manifold optimization Hu et al. (2020).

Building upon the existing theoretical foundation and analysing the advantages and disadvantages of the above optimization methods, this paper incorporates principles from machine learning, particularly

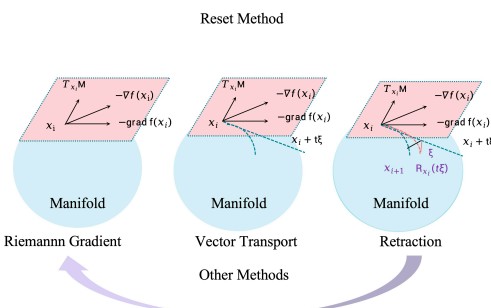

Figure 1: **An illustrative diagram for Reset method based on the theory of manifold optimization on real manifolds**. **(1) Vector Transport in the section 4.1.2.** In the Riemannian manifold, the gradient of different points is located on the tangent space of that point. To compare the Riemannian gradient at different points, for example, $\mathrm{gradf}(x_i)$ and $\mathrm{gradf}(x_{i+1})$, the solution is to transport one of them to the tangent space of the other one. **(2) Contraction Operator in the section 4.1.3.** With a negative gradient, the next step is how to go one step forward, assuming that $x_i \in M$, and the Riemannian gradient at that point: $\mathrm{gradf}(x_i)$ is known in order to obtain $x_{i+1} \in M$, this process can be implemented by adopting the theory of linear self-isomorphisms in definition 6. **(3) Reset method** For more details, see eq. (5), where a novel method for updating the gradient is proposed.

deep learning, to propose novel optimization methods aimed at enhancing the stability of model training. In simple terms, a manifold optimization algorithm perceives a constrained optimization problem in Euclidean space as an unconstrained optimization problem Hu et al. (2020) on a manifold. Similar to unconstrained optimization Hu et al. (2020) algorithms in Euclidean space, the algorithm seeks an appropriate descent direction within the tangent space Hu et al. (2020) of the current iteration point.

A restart technique is introduced by O'donoghue & Candes (2015), which can be interpreted to mean that different optimization methods are additive. Our main contribution is that the detailed analysis of the advantages and disadvantages of several existing manifold optimization methods is provided. Based on that, Reset method is proposed to improve the convergence trajectory and model stability by utilizing other methods (SGD, Adam Kingma & Ba (2014) and AdamW Loshchilov & Hutter (2017)).

## 2 RELATED WORK

This section can be found in section A in the appendix

## 3 PRELIMINARY:MANIFOLD THEORY

This section can be found in section B in the appendix.

## 4 METHODOLOGY: RESET METHOD

While optimization methods in Euclidean space have been well-established, manifold optimization Hu et al. (2020) methods are different because not all the properties that hold in Euclidean space can be directly applied to Riemannian manifolds. Therefore, it becomes necessary to redefine some properties to suit the specific characteristics of the manifold.

In this section, motivated by the restart technique introduced by O'donoghue & Candes (2015) and a novel Reset method is proposed, i.e., the manifold optimization methods are ran for a few iterations and then reset it with other methods (SGD, Adam Kingma & Ba (2014) and AdamW Loshchilov & Hutter (2017)). Our methods in algorithm 1 are described, and the convergences of our methods are proved in theorem 4.1. In addition to its practicality, the advantage of our Reset method is that our methods can improve the convergence trajectory and increase the stability of the model.

### 4.1 PRELIMINARY ON MANIFOLD OPTIMIZATION

#### 4.1.1 OBJECTIVE FUNCTION

The manifold optimization algorithm on the Riemannian manifold is to resolve the following problem:

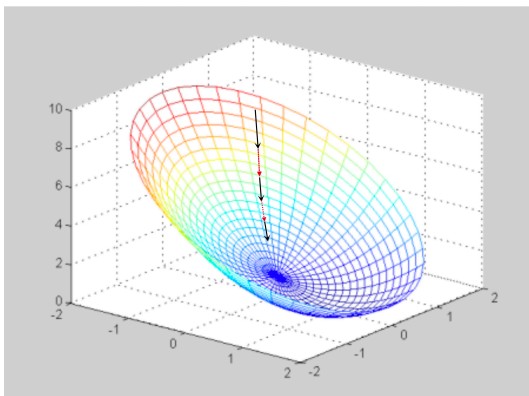

**Figure 2:** An illustrative diagram for Reset method based on the theory of manifold optimization on real manifolds. The black arrows represent the original gradients, and the red arrows represent the gradients of the Reset method, it is observed that the improvement in convergence is very powerful, and the purpose of our methods is to make the gradient approach the optimal point in the fastest way, so that the gradient update is more significant.

$$\min_{x \in \mathcal{M}} f(x)$$

$f : \mathcal{M} \to \mathbb{R}$, the function is a smooth one. $\mathcal{M}$ may be a smooth, possibly nonlinear space.

The manifold optimization is a well established optimization framework designed to solve optimization problems defined on certain nonlinear spaces. An example is presented to illustrate the theory of manifold optimization with the simplest nonlinear space:

$$\mathcal{M} = S^{n-1}, \mathcal{M} = \{x \in \mathbb{R}^n : \|x\| = 1\} \tag{1}$$

To design optimization methods for manifolds, the gradient plays a crucial role. It is essential to define the gradient on the manifold, known as the Riemannian gradient, and ensure that it is constrained to the tangent space Hu et al. (2020) of the manifold. $\exists \mathrm{grad}\mathrm{f}(\mathrm{x}) \in T_x M$, it is expressed as the unique tangent vector Hu et al. (2020), which satisfies $\langle \mathrm{grad}\mathrm{f}(\mathrm{x}), \xi \rangle_x = df(x)[\xi], \forall \xi \in T_x M$.

The tangent space Hu et al. (2020) can be considered as a linear approximation of the manifold around a specific point. By considering a sufficiently small neighbourhood, the discrepancy between the tangent space Hu et al. (2020) and the manifold can be controlled.

### 4.1.2 VECTOR TRANSFER OPERATOR

In a Riemannian manifold, the gradients at different points are located within the tangent space Hu et al. (2020) of their respective points. To compare the Riemannian gradients at two different points, such as $\mathrm{grad} f(x_i)$ and $\mathrm{grad} f(x_{i+1})$, it is necessary to "translate" one of the gradients in to the tangent space Hu et al. (2020) of the other with Vector Transport Hu et al. (2020).

**Definition 1** (Vector Transport Hu et al. (2020)). A vector transport Hu et al. (2020) is a smooth mapping on a manifold $\mathcal{M} : \mathcal{T}\mathcal{M} \oplus \mathcal{T}\mathcal{M} \to \mathcal{T}\mathcal{M} : (\xi_x, \eta_x) \rightarrowtail \mathcal{T}_{\xi_x}(\eta_x) \in \mathcal{T}\mathcal{M}$, which satisfies the following properties:

(i)There exists a retraction $\mathcal{R}$, which is linked with $\mathcal{T}$, such that $\mathcal{T}_{\eta_x}(\xi_x) \in \mathcal{T}_{\mathcal{R}_x(\xi_x)} M$.

(ii)$\mathcal{T}_{0_x}(\xi_x) = \xi_x, \forall \xi_x \in \mathcal{T}_z \mathcal{M}$.

(iii)There exists a mapping: $\mathcal{T}_{\eta_x} : \mathcal{T}_z \mathcal{M} \to \mathcal{T}_{\mathcal{R}_x(\eta_x)} \mathcal{M} : \xi_x \rightarrowtail \mathcal{T}_{\eta_x}(\xi_x)$ is linear.

### 4.1.3 CONTRACTION OPERATOR

In order to progress from a given point $x_i \in \mathcal{M}$ on a Riemannian manifold, the known Riemannian gradient $\mathrm{grad} f(x_i)$ at that point is employed, the following definitions are provided:

**Definition 2.** Contraction Operator Hu et al. (2020). $\mathcal{M}$ is a smooth mapping, $\mathcal{R} : \mathcal{T}\mathcal{M} \to \mathcal{M}$ is a retractionHu et al. (2020) on the $\mathcal{M}$, let $\mathcal{R}_z : \mathcal{T}_z \mathcal{M} \to \mathcal{M}$ represent the restriction of $\mathcal{R}$ to $\mathcal{T}_z \mathcal{M}$ :

Table 1: Manifolds and Its Abbreviation

| Real Manifolds | Abbreviation |
| --- | --- |
| Euclidean Manifolds Phogat & Chang (2022) | e |
| The Manifold of Fixed Rank Matrices Vandereycken (2013) | fr |
| Grassmann Manifold Gu et al. (2023) | g |
| The Oblique Manifold Absil & Gallivan (2006) | o |
| The Product Manifold Rovenski & Walczak (2023) | p |
| Manifold of Positive Symmetric Definite Matrices Jayasumana et al. (2013) | psd |
| Stiefel Manifold Chen et al. (2023) | s |
| The Special Orthogonal Group Mahony et al. (2008) | sog |
| The Sphere Manifolds Trendafilov (2010) | sp |
| The Manifold of Strictly Positive Vectors | spv |

(i) $\mathscr{R}_z(0_z) = z$, $0_z$ is the zero element of $\mathscr{T}_z\mathscr{M}$.

(ii) $d\mathscr{R}_z(0_z) = id_{\mathscr{T}_z\mathscr{M}}$, $id_{\mathscr{T}_z\mathscr{M}}$ is the identity mapping on $\mathscr{T}_z\mathscr{M}$.

### 4.1.4 OTHER METHODS

**Stochastic Gradient Descent (SGD)** Stochastic Gradient Descent (SGD) is a commonly used type of optimization algorithm, which is widely used for model training in machine learning and deep learning. The core idea of SGD is to use only the gradient information of one sample in each iteration to update the model parameters, the parameters are updated for each iteration with the following equation:

$$g_0 \leftarrow \frac{\partial J(\theta)}{\partial \theta}, \theta \leftarrow \theta - \eta g_0. \tag{2}$$

$J(\theta)$ denotes the loss function, $\eta$ denotes the learning rate, in general, $J(\theta)$ is not equal to $f(x)$, $f(x)$ is an objective function.

**Adaptive Moment Estimation (Adam)** Adaptive Moment Estimation (Adam) Kingma & Ba (2014) stands out as an efficient optimization algorithm. It aims to adjust the learning rate of each parameter. The Adam Kingma & Ba (2014) optimizer works by dynamically adjusting the step size, which is larger in simpler regions and smaller in more complex regions, to ensure that the minimum point, which represents the minimum loss in machine learning, is reached more efficiently and faster.

By adaptively adjusting the step size, the Adam Kingma & Ba (2014) optimizer is better able to adapt to gradient variations in different parameters and thus converge to the optimal solution faster. This strategy of dynamically adjusting the learning rate makes Adam Kingma & Ba (2014) more robust during training and overcomes the disadvantage of manually adjusting the learning rate in traditional optimization algorithms, the parameters are updated for each iteration with the following equation:

$$v_t = \alpha v_{t-1} + (1 - \alpha)\frac{\partial J(\theta)}{\partial \theta}, \theta \leftarrow \theta - \eta v_t. \tag{3}$$

$J(\theta)$ denotes the loss function, $\eta$ denotes the learning rate, $v_t$ refers to momentum, in general, $J(\theta)$ is not equal to $f(x)$, $f(x)$ is an objective function.

**Adam with Weight Decay Fix (AdamW)** The AdamW Loshchilov & Hutter (2017) optimizer is a variant of the Adam Kingma & Ba (2014) optimizer that combines weight decay (L2 regularization) with the Adam Kingma & Ba (2014) optimizer. The key to AdamW Loshchilov & Hutter (2017) is that it treats weight decay separately from gradient updating, which helps to address the incompatibility of L2 regularization with adaptive learning rate algorithms(such as Adam Kingma & Ba (2014)), the parameters are updated for each iteration with the following equation:

$$v_t = \alpha v_{t-1} + (1 - \alpha)\frac{\partial J(\theta)}{\partial \theta}, \theta \leftarrow \theta - \eta v_t, \theta \leftarrow \theta - \lambda \eta v_t. \tag{4}$$

$J(\theta)$ denotes the loss function, $\eta$ denotes the learning rate, $v_t$ refers to momentum, $\lambda$ refers to the regularization parameter, in general, $J(\theta)$ is not equal to $f(x)$, $f(x)$ is an objective function.

### 4.2 PROPOSED METHODS

#### 4.2.1 RESET METHOD ON THE REAL MANIFOLDS

To enhance the stability of the optimizer, Reset method is proposed to cooperate optimization process. The idea is to introduce a step size correction function, denoted as $B_{x_i}$, that operates on the step sizes $x_i$ and $x_{i+1}$, which is then used to compute $x_{i+2}$ in the optimizer iteration.

The specific calculation steps of Reset method on the the real manifolds are given below:

$$\begin{cases} x_i = R_{x_{i-1}}(-\alpha_{i-1}\mathrm{gradf}(x_{i-1})) \\ x_{i+1} = B_{x_i}(x_i) \\ x_{i+2} = R_{x_{i+1}}(-\alpha_{i+1}\mathrm{gradf}(x_{i+1})) \end{cases} \tag{5}$$

$J(\theta)$ denotes the loss function.

Regarding the selection of the function $B_{x_i}$, this paper selects the methods (SGD, Adam Kingma & Ba (2014) and AdamW Loshchilov & Hutter (2017)). Regarding the selection of the step size, the step size factor is a constant parameter to be determined in order to control the stability of the Reset method and the convergence speed of the Reset method.

In contrast to the linear search method commonly used in Euclidean space, the step size determination on a manifold involves conducting a curve search.

In optimization theory, there is a classical criterion for line search algorithms, namely the Armijo criterion. The definition of the Armijo criterion in two steps is given:

(i) Just set $\phi \in (0,1), \psi \in (0,0.5)$, the definition of the step size factor is given, and let the step size factor be defined as:

$$\psi_k = \phi^{\lambda_i}$$

(ii) $\lambda_i$ is a non-negative integer and is the smallest one, $d_i$ is the search direction vector and $g_i$ is the gradient. $\lambda_i$ satisfies the following inequality:

$$f(x_i + \phi^{\lambda_i}d_i) \le f(x_i) + \psi\phi^{\lambda_i}g_i^T d_i$$

Based upon the definitions above, the Armijo search is taken as an example, given $\alpha, \epsilon \in (0,1)$, the smallest non-negative integer $\epsilon_0$ is that:

$$f(R_{x_{i-1}}(-\alpha_{i-1}\mathrm{gradf}(x_{i-1}))) \le f(x_{i-1}) + \alpha(-\alpha_{i-1})\langle \mathrm{gradf}(x_{i-1}), \mathrm{gradf}(x_{i-1})\rangle_{x_{i-1}}$$
$$f(B_{x_i}(x_i)) \le f(x_i) + \alpha\langle \mathrm{gradf}(x_i), x_i\rangle_{x_i} \tag{6}$$
$$f(R_{x_{i+1}}(-\alpha_{i+1}\mathrm{gradf}(x_{i+1}))) \le f(x_{i+1}) + \alpha(-\alpha_{i+1})\langle \mathrm{gradf}(x_{i+1}), \mathrm{gradf}(x_{i+1})\rangle_{x_{i+1}}$$

$$f(R_{x_{i-1}}(-\alpha_{i-1}\mathrm{gradf}(x_{i-1}))) \le C_{i-1} + \alpha(-\alpha_{i-1})\langle \mathrm{gradf}(x_{i-1}), \mathrm{gradf}(x_{i-1})\rangle_{x_{i-1}}$$
$$f(B_{x_i}(x_i)) \le C_i + \alpha\langle \mathrm{gradf}(x_i), x_i\rangle_{x_i} \tag{7}$$
$$f(R_{x_{i+1}}(-\alpha_{i+1}\mathrm{gradf}(x_{i+1}))) \le C_{i+1} + \alpha(-\alpha_{i+1})\langle \mathrm{gradf}(x_{i+1}), \mathrm{gradf}(x_{i+1})\rangle_{x_{i+1}}$$

where $-\alpha_{i-1} = \beta_i\epsilon^{\epsilon_0}$ and $\beta_i$ is the initial step size. $C_{i-1}$ is a convex combination of $C_{i-1}$ and $f(x_{i-1})$ via $C_{i-1} = (\zeta R_{i-2}C_{i-2} + f(x_{i-1}))/R_{i-1}$, where $\zeta \in [0,1], C_0 = f(x_0), R_{i+1} = \zeta R_i + 1$ and $R_0 = 1$. The Barzilai-Borwein (BB) method is a gradient descent method, in order to avoid the computational complexity that comes from computing the second order derivatives, and with this, this method can speed up the convergence rate. The Barzilai-Borwein (BB) method may be generalized to Riemannian manifold as:

$$\zeta_{i_1} = \frac{\langle \omega_{i-1}, \omega_{i-1}\rangle_{x_i}}{|\langle \omega_{i-1}, v_{i-1}\rangle_{x_i}|}, \zeta_{i_2} = \frac{|\langle \omega_{i-1}, v_{i-1}\rangle_{x_i}|}{\langle \omega_{i-1}, v_{i-1}\rangle_{x_i}} \tag{8}$$

where

$$\omega_{i-1} = \alpha_{i-1}\mathscr{T}_{x_{i-1}\to x_i}(\mathrm{gradf}(x_{i-1})), v_{i-1} = \mathrm{gradf}(x_i) - \alpha_{i-1}^{-1}\omega_{i-1}. \tag{9}$$

### 4.2.2 Theorems of Reset Method on the Real Manifolds

As for convergence, known from (Hu et al. (2020)) that the step size of manifold optimization is convergent, so whether or not the method converges after improving it, and whether there is a logically clear proof, theorem 4.1 returns to this question.

**Theorem 4.1.** Let $x_i$ be the sequence generated by Reset method on the real manifolds using non-monotonic sequences, it is assumed that $f$ is continuously differential on the real manifold $\mathscr{M}$ and Euclidean space $\mathscr{E}$. In this context, every accumulation point $x^*$ of the sequence $x_i$ is considered a stationary point of the optimization problem, i.e., it holds $\mathrm{gradf}(x^*) = 0$.

---

**Algorithm 1:** Reset Method

---

**Input:** $x_0 \in \mathcal{M}$. Set $\zeta_{max} \in [1, +\infty)$, $\zeta_{min} \in [0,1]$, $R_0 = 1$, $C_0 = f(x_0)$.
**while:** $\mathrm{gradf}(x_i) \neq 0$ **do**
**Compute** $\zeta_i$**, according to eq. (7) and set**
$\zeta_{i_1} = max(\zeta_{min}, min(\zeta_i, \zeta_{max}))$ **or** $\zeta_{i_2} = max(\zeta_{min}, min(\zeta_i, \zeta_{max}))$**.**
**Then, calculate** $R_i$**,** $C_i$ **and seek a step size** $\alpha_i$ **contenting eq. (6).**
**Set** $B_{x_i}(x_i) \leftarrow R_{x_{i-1}}(-\alpha_{i-1}\mathrm{gradf}(x_{i-1}))$.
**Set** $i - 1 \leftarrow i$.
**Set** $R_{x_{i+1}}(-\alpha_{i+1}\mathrm{gradf}(x_{i+1})) \leftarrow B_{x_i}(x_i)$.
**Set** $i \leftarrow i + 1$.

---

The proof process can be found in section C.1.1 in the appendix.

It follows from the theorem 4.1 that this stabilization point exists and the Reset method may have an error compared with the common optimization method, the Reset method improves convergence since it can produce damped harmonic motion Yadav et al. (2018) that sinks into the saddle point. From our knowledge of the dynamical system Yadav et al. (2018), it is understand that the damping produced by the Reset method causes the orbit to converge to the saddle point and the error decays at an exponential rate.

The above is from a mathematical point of view of the dynamical system Yadav et al. (2018) to explain that our methods make a more appropriate choice, but it may cause errors. So we need to calculate this error, and this error should be as small as possible, the smaller the error, the smaller the amount of loss(cost), which is the desired result. In fact, the error is very small, the theorem 4.2 explains this in detail.

**Theorem 4.2.** Let $x_i$ be the sequence generated by Reset method on the real manifolds using a non-monotonic sequence. It is assumed that the objective function $f$ is continuously differentiable on the real manifold $\mathcal{M}$. Every accumulation point $x^*$ of the sequence $x_i$ is considered a stationary point of the optimization problem. Let $x^\star$, $x^\heartsuit$ and $x^\diamond$ be a point obtained after backward propagation of the gradient using other methods (SGD, Adam Kingma & Ba (2014) and AdamW Loshchilov & Hutter (2017)) respectively. Furthermore, it is assumed that there exists a stochastic gradient which satisfies $\mathbb{E}[\|\mathrm{gradf}(x_i^*)\|^2] \leq \epsilon_0^2$, $\mathbb{E}[\|\mathrm{gradf}(x_i^\star)\|^2] \leq \epsilon_1^2$, $\mathbb{E}[\|\mathrm{gradf}(x_i^\heartsuit)\|^2] \leq \epsilon_2^2$, $\mathbb{E}[\|\mathrm{gradf}(x_i^\diamond)\|^2] \leq \epsilon_3^2$, the error bound is that:

$$\mathbb{E}[\mathrm{gradf}(x_{i+1}^\star)] - \mathbb{E}[\mathrm{gradf}(x_i)] \leq \frac{1}{2}(\frac{4i^2}{(i+1)^2}\epsilon_1^2 + \epsilon_0^2),$$

$$\mathbb{E}[\mathrm{gradf}(x_{i+1}^\heartsuit)] - \mathbb{E}[\mathrm{gradf}(x_i)] \leq \frac{1}{2}(\frac{4i^2}{(i+1)^2}\epsilon_2^2 + \epsilon_0^2), \qquad (10)$$

$$\mathbb{E}[\mathrm{gradf}(x_{i+1}^\diamond)] - \mathbb{E}[\mathrm{gradf}(x_i)] \leq \frac{1}{2}(\frac{4i^2}{(i+1)^2}\epsilon_3^2 + \epsilon_0^2).$$

The process of prove can be found in section C.1.1 in the Appendix.

# 5 DEEP LEARNING EXPERIMENTS

## 5.1 DATASETS

Please review this section in section D.1 of the Appendix.

## 5.2 DEEP LEARNING EXPERIMENTS

### 5.2.1 STABLE GENERATIVE ADVERSARIAL NETWORK (STABLEGAN) FOR IMAGE GENERATION

The CIFAR-100 Krizhevsky et al. (2009) dataset, the STL-10 dataset, the SVHN (Street View House Numbers) dataset and the CIFAR-10 Krizhevsky et al. (2009) dataset are picked up as a measure of image generation performance. For our methods, it is evaluated at the Stable Generative Adversarial

Table 2: Image Generation of the Deep Convolutional Generative Adversarial Network (DCGAN) Yadav et al. (2017) on CIFAR-10 Krizhevsky et al. (2009) dataset

| Methods | Average Precision(AP) |
|---|---|
| Adam Kingma & Ba (2014) | 98.61 |
| Adam Kingma & Ba (2014) + e(Ours) | 99.30 |
| Adam Kingma & Ba (2014) + fr(Ours) | 99.31 |
| Adam Kingma & Ba (2014) + g(Ours) | 99.42 |
| Adam Kingma & Ba (2014) + o(Ours) | 99.46 |
| Adam Kingma & Ba (2014) + p(Ours) | 99.69 |
| Adam Kingma & Ba (2014) + psd(Ours) | 99.58 |
| Adam Kingma & Ba (2014) + s (Ours) | 99.61 |
| Adam Kingma & Ba (2014) + sog (Ours) | 99.65 |
| Adam Kingma & Ba (2014) + sp (Ours) | 99.68 |
| Adam Kingma & Ba (2014) + spv(Ours) | **99.72** |
| AdamW Loshchilov & Hutter (2017) | 98.49 |
| AdamW Loshchilov & Hutter (2017) + e (Ours) | 99.52 |
| AdamW Loshchilov & Hutter (2017) + fr (Ours) | 99.61 |
| AdamW Loshchilov & Hutter (2017) + g (Ours) | 99.75 |
| AdamW Loshchilov & Hutter (2017) + o (Ours) | 99.63 |
| AdamW Loshchilov & Hutter (2017) + p (Ours) | 99.75 |
| AdamW Loshchilov & Hutter (2017) + psd (Ours) | 99.62 |
| AdamW Loshchilov & Hutter (2017) + s (Ours) | 99.75 |
| AdamW Loshchilov & Hutter (2017) + sog (Ours) | 99.64 |
| AdamW Loshchilov & Hutter (2017) + sp(Ours) | 99.76 |
| AdamW Loshchilov & Hutter (2017) + spv(Ours) | **99.78** |
| SGD | 98.20 |
| SGD + e (Ours) | 98.62 |
| SGD + fr (Ours) | 98.56 |
| SGD + g (Ours) | 98.63 |
| SGD + o (Ours) | 98.59 |
| SGD + p (Ours) | 98.74 |
| SGD + psd (Ours) | 98.68 |
| SGD + s (Ours) | 98.72 |
| SGD + sog (Ours) | 98.74 |
| SGD + sp (Ours) | 98.75 |
| SGD + spv (Ours) | **98.70** |

Table 3: Image Generation of the Deep Convolutional Generative Adversarial Network (DCGAN) Yadav et al. (2017) on CIFAR-100 Krizhevsky et al. (2009) dataset

| Methods | Average Precision(AP) |
|---|---|
| Adam Kingma & Ba (2014) | 98.59 |
| Adam Kingma & Ba (2014) + e(Ours) | 99.82 |
| Adam Kingma & Ba (2014) + fr(Ours) | 99.31 |
| Adam Kingma & Ba (2014) + g(Ours) | 99.42 |
| Adam Kingma & Ba (2014) + o(Ours) | 99.46 |
| Adam Kingma & Ba (2014) + p(Ours) | 99.51 |
| Adam Kingma & Ba (2014) + psd(Ours) | 99.58 |
| Adam Kingma & Ba (2014) + s (Ours) | 99.61 |
| Adam Kingma & Ba (2014) + sog (Ours) | 99.65 |
| Adam Kingma & Ba (2014) + sp (Ours) | 99.68 |
| Adam Kingma & Ba (2014) + spv(Ours) | **99.72** |
| AdamW Loshchilov & Hutter (2017) | 98.61 |
| AdamW Loshchilov & Hutter (2017) + e (Ours) | 99.56 |
| AdamW Loshchilov & Hutter (2017) + fr (Ours) | 99.61 |
| AdamW Loshchilov & Hutter (2017) + g (Ours) | 99.75 |
| AdamW Loshchilov & Hutter (2017) + o (Ours) | 99.86 |
| AdamW Loshchilov & Hutter (2017) + p (Ours) | 99.95 |
| AdamW Loshchilov & Hutter (2017) + psd (Ours) | 99.25 |
| AdamW Loshchilov & Hutter (2017) + s (Ours) | 99.38 |
| AdamW Loshchilov & Hutter (2017) + sog (Ours) | 99.44 |
| AdamW Loshchilov & Hutter (2017) + sp(Ours) | 99.56 |
| AdamW Loshchilov & Hutter (2017) + spv(Ours) | **99.79** |
| SGD | 98.24 |
| SGD + e (Ours) | 99.32 |
| SGD + fr (Ours) | 99.36 |
| SGD + g (Ours) | 99.43 |
| SGD + o (Ours) | 99.49 |
| SGD + p (Ours) | 99.53 |
| SGD + psd (Ours) | 99.58 |
| SGD + s (Ours) | 99.63 |
| SGD + sog (Ours) | 99.68 |
| SGD + sp (Ours) | 99.72 |
| SGD + spv (Ours) | **99.76** |

Network (StableGAN) Yadav et al. (2018) with an initial learning rate of 0.002, and the StableGAN Yadav et al. (2018) is trained with 10,000 iterations on 4V100 GPUs at a scale of 64 batches. Through experiments, the results confirm our intuition and validated the effectiveness and stability of our methods. The average precision performance is summarized in table 4,table 5,table 6 and table 7, our methods show a higher average precision than the competitors.

### 5.2.2 DEEP CONVOLUTIONAL GENERATIVE ADVERSARIAL NETWORK (DCGAN) FOR IMAGE GENERATION

The CIFAR-10 Krizhevsky et al. (2009) dataset, the STL-10 dataset, the SVHN (Street View House Numbers) dataset and the CIFAR-100 Krizhevsky et al. (2009) dataset are selected as a measure of domain adaptation performance. Our methods are evaluated on the Deep Convolutional Generative Adversarial Network (DCGAN) Yadav et al. (2017) backbone with an initial learning rate of 0.0002, and the model is trained with 5,000 iterations on 4 V100 GPUs at a scale of 32 batches and the learning rate warmup He et al. (2019) is employed.

Table 4: Image Generation of the Stable Generative Adversarial Network (StableGAN) Yadav et al. (2018) on CIFAR-10 Krizhevsky et al. (2009) dataset

| Methods | Average Precision(AP) |
|---|---|
| Adam Kingma & Ba (2014) | 98.59 |
| Adam Kingma & Ba (2014) + e(Ours) | 99.30 |
| Adam Kingma & Ba (2014) + fr(Ours) | 99.31 |
| Adam Kingma & Ba (2014) + g(Ours) | 99.42 |
| Adam Kingma & Ba (2014) + o(Ours) | 99.46 |
| Adam Kingma & Ba (2014) + p(Ours) | 99.69 |
| Adam Kingma & Ba (2014) + psd(Ours) | 99.58 |
| Adam Kingma & Ba (2014) + s (Ours) | 99.61 |
| Adam Kingma & Ba (2014) + sog (Ours) | 99.65 |
| Adam Kingma & Ba (2014) + sp (Ours) | 99.68 |
| Adam Kingma & Ba (2014) + spv(Ours) | **99.70** |
| AdamW Loshchilov & Hutter (2017) | 98.52 |
| AdamW Loshchilov & Hutter (2017) + e (Ours) | 99.52 |
| AdamW Loshchilov & Hutter (2017) + fr (Ours) | 99.61 |
| AdamW Loshchilov & Hutter (2017) + g (Ours) | 99.75 |
| AdamW Loshchilov & Hutter (2017) + o (Ours) | 99.63 |
| AdamW Loshchilov & Hutter (2017) + p (Ours) | 99.75 |
| AdamW Loshchilov & Hutter (2017) + psd (Ours) | 99.62 |
| AdamW Loshchilov & Hutter (2017) + s (Ours) | 99.75 |
| AdamW Loshchilov & Hutter (2017) + sog (Ours) | 99.64 |
| AdamW Loshchilov & Hutter (2017) + sp(Ours) | 99.76 |
| AdamW Loshchilov & Hutter (2017) + spv(Ours) | **99.78** |
| SGD | 98.31 |
| SGD + e (Ours) | 98.62 |
| SGD + fr (Ours) | 98.56 |
| SGD + g (Ours) | 98.63 |
| SGD + o (Ours) | 98.59 |
| SGD + p (Ours) | 98.64 |
| SGD + psd (Ours) | 98.68 |
| SGD + s (Ours) | 98.62 |
| SGD + sog (Ours) | 98.64 |
| SGD + sp (Ours) | 98.65 |
| SGD + spv (Ours) | **98.68** |

Table 5: Image Generation of the Stable Generative Adversarial Network (StableGAN) Yadav et al. (2018) on CIFAR-100 Krizhevsky et al. (2009) dataset

| Methods | Average Precision(AP) |
|---|---|
| Adam Kingma & Ba (2014) | 98.71 |
| Adam Kingma & Ba (2014) + e(Ours) | 99.81 |
| Adam Kingma & Ba (2014) + fr(Ours) | 99.79 |
| Adam Kingma & Ba (2014) + g(Ours) | 99.80 |
| Adam Kingma & Ba (2014) + o(Ours) | 99.78 |
| Adam Kingma & Ba (2014) + p(Ours) | 99.81 |
| Adam Kingma & Ba (2014) + psd(Ours) | 99.76 |
| Adam Kingma & Ba (2014) + s (Ours) | 99.78 |
| Adam Kingma & Ba (2014) + sog (Ours) | 99.77 |
| Adam Kingma & Ba (2014) + sp (Ours) | 99.79 |
| Adam Kingma & Ba (2014) + spv(Ours) | **99.82** |
| AdamW Loshchilov & Hutter (2017) | 98.51 |
| AdamW Loshchilov & Hutter (2017) + e (Ours) | 99.82 |
| AdamW Loshchilov & Hutter (2017) + fr (Ours) | 99.81 |
| AdamW Loshchilov & Hutter (2017) + g (Ours) | 99.85 |
| AdamW Loshchilov & Hutter (2017) + o (Ours) | 99.83 |
| AdamW Loshchilov & Hutter (2017) + p (Ours) | 99.81 |
| AdamW Loshchilov & Hutter (2017) + psd (Ours) | 99.85 |
| AdamW Loshchilov & Hutter (2017) + s (Ours) | 99.78 |
| AdamW Loshchilov & Hutter (2017) + sog (Ours) | 99.84 |
| AdamW Loshchilov & Hutter (2017) + sp(Ours) | 99.86 |
| AdamW Loshchilov & Hutter (2017) + spv(Ours) | **99.86** |
| SGD | 98.36 |
| SGD + e (Ours) | 99.72 |
| SGD + fr (Ours) | 99.76 |
| SGD + g (Ours) | 99.73 |
| SGD + o (Ours) | 99.69 |
| SGD + p (Ours) | 99.73 |
| SGD + psd (Ours) | 99.72 |
| SGD + s (Ours) | 99.73 |
| SGD + sog (Ours) | 99.75 |
| SGD + sp (Ours) | 99.72 |
| SGD + spv (Ours) | **99.79** |

Through experiments, the results confirm our intuition and validated the effectiveness and stability of our methods. The average precision performance is summarized in table 2, table 3, table 10 and table 11, from the table, it is known that our methods are more stable and shows a higher average precision than the competitors.

### 5.2.3 CLUSTER CONTRAST FOR UNSUPERVISED PERSON RE-IDENTIFICATION

The DukeMTMC-reID Zheng et al. (2017) dataset and Market1501 Zheng et al. (2015) dataset are selected as a measure of unsupervised learning tasks for object re-ID and person re-ID. Our methods are evaluated on the improved ResNet50 called Cluster Contrast Dai et al. (2022) backbone with an

Table 6: Image Generation of the Stable Generative Adversarial Network (StableGAN) Yadav et al. (2018) on STL-10 dataset

| Methods | Average Precision(AP) |
| --- | --- |
| Adam Kingma & Ba (2014) | 98.59 |
| Adam Kingma & Ba (2014) + e(Ours) | 99.56 |
| Adam Kingma & Ba (2014) + fr(Ours) | 99.58 |
| Adam Kingma & Ba (2014) + g(Ours) | 99.59 |
| Adam Kingma & Ba (2014) + o(Ours) | 99.61 |
| Adam Kingma & Ba (2014) + p(Ours) | 99.59 |
| Adam Kingma & Ba (2014) + psd(Ours) | 99.62 |
| Adam Kingma & Ba (2014) + s (Ours) | 99.58 |
| Adam Kingma & Ba (2014) + sog (Ours) | 99.62 |
| Adam Kingma & Ba (2014) + sp (Ours) | 99.65 |
| Adam Kingma & Ba (2014) + spv(Ours) | **99.71** |
| AdamW Loshchilov & Hutter (2017) | 98.52 |
| AdamW Loshchilov & Hutter (2017) + e (Ours) | 99.56 |
| AdamW Loshchilov & Hutter (2017) + fr (Ours) | 99.61 |
| AdamW Loshchilov & Hutter (2017) + g (Ours) | 99.75 |
| AdamW Loshchilov & Hutter (2017) + o (Ours) | 99.65 |
| AdamW Loshchilov & Hutter (2017) + p (Ours) | 99.75 |
| AdamW Loshchilov & Hutter (2017) + psd (Ours) | 99.72 |
| AdamW Loshchilov & Hutter (2017) + s (Ours) | 99.68 |
| AdamW Loshchilov & Hutter (2017) + sog (Ours) | 99.74 |
| AdamW Loshchilov & Hutter (2017) + sp(Ours) | 99.76 |
| AdamW Loshchilov & Hutter (2017) + spv(Ours) | **99.78** |
| SGD | 98.28 |
| SGD + e (Ours) | 99.62 |
| SGD + fr (Ours) | 99.66 |
| SGD + g (Ours) | 99.63 |
| SGD + o (Ours) | 99.69 |
| SGD + p (Ours) | 99.63 |
| SGD + psd (Ours) | 99.68 |
| SGD + s (Ours) | 99.63 |
| SGD + sog (Ours) | 99.68 |
| SGD + sp (Ours) | 99.62 |
| SGD + spv (Ours) | **98.69** |

Table 7: Image Generation of the Stable Generative Adversarial Network (StableGAN) Yadav et al. (2018) on SVHN dataset

| Methods | Average Precision(AP) |
| --- | --- |
| Adam Kingma & Ba (2014) | 98.63 |
| Adam Kingma & Ba (2014) + e(Ours) | 99.80 |
| Adam Kingma & Ba (2014) + fr(Ours) | 99.81 |
| Adam Kingma & Ba (2014) + g(Ours) | 99.82 |
| Adam Kingma & Ba (2014) + o(Ours) | 99.76 |
| Adam Kingma & Ba (2014) + p(Ours) | 99.81 |
| Adam Kingma & Ba (2014) + psd(Ours) | 99.78 |
| Adam Kingma & Ba (2014) + s (Ours) | 99.81 |
| Adam Kingma & Ba (2014) + sog (Ours) | 99.77 |
| Adam Kingma & Ba (2014) + sp (Ours) | 99.78 |
| Adam Kingma & Ba (2014) + spv(Ours) | **99.82** |
| AdamW Loshchilov & Hutter (2017) | 98.51 |
| AdamW Loshchilov & Hutter (2017) + e (Ours) | 99.86 |
| AdamW Loshchilov & Hutter (2017) + fr (Ours) | 99.81 |
| AdamW Loshchilov & Hutter (2017) + g (Ours) | 99.85 |
| AdamW Loshchilov & Hutter (2017) + o (Ours) | 99.86 |
| AdamW Loshchilov & Hutter (2017) + p (Ours) | 99.85 |
| AdamW Loshchilov & Hutter (2017) + psd (Ours) | 99.82 |
| AdamW Loshchilov & Hutter (2017) + s (Ours) | 99.88 |
| AdamW Loshchilov & Hutter (2017) + sog (Ours) | 99.84 |
| AdamW Loshchilov & Hutter (2017) + sp(Ours) | 99.86 |
| AdamW Loshchilov & Hutter (2017) + spv(Ours) | **99.88** |
| SGD | 98.26 |
| SGD + e (Ours) | 99.72 |
| SGD + fr (Ours) | 99.76 |
| SGD + g (Ours) | 99.73 |
| SGD + o (Ours) | 99.79 |
| SGD + p (Ours) | 99.73 |
| SGD + psd (Ours) | 99.76 |
| SGD + s (Ours) | 99.72 |
| SGD + sog (Ours) | 99.78 |
| SGD + sp (Ours) | 99.72 |
| SGD + spv (Ours) | **99.80** |

initial learning rate of 0.001, and the model is trained with 400 iterations on 4 V100 GPUs at a scale of 256 batches and the learning rate warmup He et al. (2019) is employed.

Through experiments, the results confirm our intuition and validated the effectiveness and stability of our methods. The average precision performance is summarized in table 8 and table 9, from the table, it is known that our methods are more stable and shows a higher average precision than the competitors.

## 6 CONCLUSION

In this paper, the Reset method is introduced that combines other methods (SGD,Adam Kingma & Ba (2014) and AdamW Loshchilov & Hutter (2017)) to address the challenges of convergence loss and

Table 8: Cluster Contrast Dai et al. (2022) for Unsupervised Person Re-Identification on the DukeMTMC-reID Zheng et al. (2017) dataset

| Datasets | DukeMTMC-reID Zheng et al. (2017) dataset | | | |
|---|---|---|---|---|
| Methods | mAP | top-1 | top-5 | top-10 |
| Adam Kingma & Ba (2014) | 73.6 | 84.5 | 90.2 | 92.9 |
| Adam Kingma & Ba (2014) + e(Ours) | **82.6** | **86.6** | **92.2** | **94.0** |
| Adam Kingma & Ba (2014) + fr(Ours) | 81.9 | 85.7 | 91.4 | 93.5 |
| Adam Kingma & Ba (2014) + g(Ours) | 81.9 | 85.9 | 91.5 | 93.6 |
| Adam Kingma & Ba (2014) + o(Ours) | 82.0 | 85.6 | 91.2 | 93.3 |
| Adam Kingma & Ba (2014) + p(Ours) | 80.3 | 84.6 | 90.6 | 93.0 |
| Adam Kingma & Ba (2014) + psd(Ours) | 82.5 | 86.4 | 92.0 | 93.8 |
| Adam Kingma & Ba (2014) + s (Ours) | 81.6 | 85.7 | 91.6 | 93.2 |
| Adam Kingma & Ba (2014) + sog (Ours) | 82.0 | 85.4 | 91.5 | 93.8 |
| Adam Kingma & Ba (2014) + sp (Ours) | 81.9 | 85.9 | 91.6 | 93.8 |
| Adam Kingma & Ba (2014) + spv(Ours) | 82.1 | 85.6 | 91.4 | 93.6 |
| AdamW Loshchilov & Hutter (2017) | 74.1 | 85.2 | 90.7 | 93.1 |
| AdamW Loshchilov & Hutter (2017) + e (Ours) | 81.6 | 85.8 | 91.7 | 93.7 |
| AdamW Loshchilov & Hutter (2017) + fr (Ours) | **82.9** | 86.2 | 91.9 | 93.7 |
| AdamW Loshchilov & Hutter (2017) + g (Ours) | 81.4 | 85.8 | 91.3 | 93.2 |
| AdamW Loshchilov & Hutter (2017) + o (Ours) | 81.3 | 85.3 | 91.4 | 93.3 |
| AdamW Loshchilov & Hutter (2017) + p (Ours) | 81.3 | 85.1 | 91.2 | 93.4 |
| AdamW Loshchilov & Hutter (2017) + psd (Ours) | 81.3 | 85.3 | 91.4 | 93.3 |
| AdamW Loshchilov & Hutter (2017) + s (Ours) | 82.3 | 86.1 | 91.7 | 93.8 |
| AdamW Loshchilov & Hutter (2017) + sog (Ours) | 81.9 | 86.0 | **92.0** | **93.9** |
| AdamW Loshchilov & Hutter (2017) + sp(Ours) | 81.4 | 85.8 | 91.0 | 93.3 |
| AdamW Loshchilov & Hutter (2017) + spv(Ours) | 82.5 | **86.5** | 91.7 | 93.6 |
| SGD | 54.9 | 65.9 | 74.6 | 80.5 |
| SGD + e (Ours) | 57.3 | 67.9 | 76.3 | 80.0 |
| SGD + fr (Ours) | 56.8 | 68.0 | 75.5 | 79.2 |
| SGD + g (Ours) | 57.4 | 67.7 | 75.8 | 79.9 |
| SGD + o (Ours) | 57.0 | 67.4 | 75.6 | 79.8 |
| SGD + p (Ours) | 57.5 | 67.7 | 76.1 | 80.1 |
| SGD + psd (Ours) | 56.7 | 67.5 | 75.6 | 79.5 |
| SGD + s (Ours) | **63.5** | **74.3** | 82.3 | **85.7** |
| SGD + sog (Ours) | 62.3 | 73.3 | 81.1 | 85.1 |
| SGD + sp (Ours) | 63.4 | 74.6 | **82.5** | 85.9 |
| SGD + spv (Ours) | 62.6 | 74.0 | 81.9 | 85.6 |

Table 9: Cluster Contrast Dai et al. (2022) for Unsupervised Person Re-Identification on the Market-1501 Zheng et al. (2015) dataset

| Datasets | Market-1501 Zheng et al. (2015) dataset | | | |
|---|---|---|---|---|
| Methods | mAP | top-1 | top-5 | top-10 |
| Adam Kingma & Ba (2014) | 83.0 | 90.1 | 93.2 | 95.0 |
| Adam Kingma & Ba (2014) + e(Ours) | 84.8 | 90.5 | 93.8 | 95.2 |
| Adam Kingma & Ba (2014) + fr(Ours) | 86.2 | 90.9 | 94.5 | 95.4 |
| Adam Kingma & Ba (2014) + g(Ours) | 83.4 | 89.5 | 93.0 | 94.6 |
| Adam Kingma & Ba (2014) + o(Ours) | 82.8 | 89.4 | 93.1 | 94.5 |
| Adam Kingma & Ba (2014) + p(Ours) | 85.7 | 90.3 | 93.8 | 95.1 |
| Adam Kingma & Ba (2014) + psd(Ours) | 84.8 | 90.1 | 93.9 | 95.5 |
| Adam Kingma & Ba (2014) + s (Ours) | 84.5 | 90.3 | 93.8 | 95.3 |
| Adam Kingma & Ba (2014) + sog (Ours) | 85.9 | 90.5 | 94.0 | 95.2 |
| Adam Kingma & Ba (2014) + sp (Ours) | **86.3** | **90.9** | **94.8** | **95.5** |
| Adam Kingma & Ba (2014) + spv(Ours) | 84.8 | 90.1 | 93.6 | 95.0 |
| AdamW Loshchilov & Hutter (2017) | 83.2 | 90.4 | 93.7 | 95.1 |
| AdamW Loshchilov & Hutter (2017) + e (Ours) | **86.5** | **90.7** | **94.2** | 95.4 |
| AdamW Loshchilov & Hutter (2017) + fr (Ours) | 85.6 | 90.2 | 94.1 | **95.6** |
| AdamW Loshchilov & Hutter (2017) + g (Ours) | 84.3 | 90.1 | 94.3 | 95.3 |
| AdamW Loshchilov & Hutter (2017) + o (Ours) | 84.7 | 90.6 | 94.0 | 94.9 |
| AdamW Loshchilov & Hutter (2017) + p (Ours) | 85.0 | 89.8 | 93.7 | 95.2 |
| AdamW Loshchilov & Hutter (2017) + psd (Ours) | 85.6 | 90.4 | 93.9 | 95.3 |
| AdamW Loshchilov & Hutter (2017) + s (Ours) | 86.4 | 90.6 | 94.1 | 95.3 |
| AdamW Loshchilov & Hutter (2017) + sog (Ours) | 85.3 | 89.9 | 93.6 | 94.9 |
| AdamW Loshchilov & Hutter (2017) + sp(Ours) | 83.5 | 89.7 | 93.6 | 94.7 |
| AdamW Loshchilov & Hutter (2017) + spv(Ours) | 86.0 | 90.1 | 93.9 | 94.9 |
| SGD | 60.8 | 72.7 | 80.5 | 83.8 |
| SGD + e (Ours) | 62.1 | 73.3 | 81.5 | 85.5 |
| SGD + fr (Ours) | 62.5 | 73.9 | 82.1 | **86.2** |
| SGD + g (Ours) | 63.2 | 74.0 | 82.0 | 85.8 |
| SGD + o (Ours) | 63.4 | 74.0 | 82.6 | 86.1 |
| SGD + p (Ours) | 63.2 | 74.0 | 82.0 | 85.8 |
| SGD + psd (Ours) | **63.5** | **74.1** | **82.7** | 86.1 |
| SGD + s (Ours) | 62.2 | 73.5 | 80.9 | 84.6 |
| SGD + sog (Ours) | 62.9 | 73.9 | 81.8 | 85.1 |
| SGD + sp (Ours) | 61.0 | 72.8 | 80.8 | 84.1 |
| SGD + spv (Ours) | 61.7 | 73.8 | 81.6 | 84.9 |

convergence speed in manifold optimization problems. Through a series of experiments, the results validate our initial intuition and confirmed the correctness and stability of our proposed method. The experiments demonstrate that Reset method combining other methods (SGD, Adam Kingma & Ba (2014) and AdamW Loshchilov & Hutter (2017)) can effectively mitigate convergence loss and improve convergence speed. These findings not only contribute to the field of optimization but also have practical implications for various machine learning applications. The Reset method combining other methods (SGD, Adam Kingma & Ba (2014) and AdamW Loshchilov & Hutter (2017)), offers a promising avenue for accelerating convergence, and improves the overall performance of optimization algorithms.

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

APPENDIX

## A RELATED WORK

### A.1 OPTIMIZATION THEORY

#### A.1.1 CONVEX OPTIMIZATION-BASED THEORY

The paper Ergen et al. (2022) shares similarities with Ergen & Pilanci (2021) in that they all combine the theory of critical normalization with convex optimization. However, they approach this combination from different perspectives, leading to distinct results. Additionally, convex pair-based analysis frameworks have been proposed in the Ergen et al. (2022), Suggala et al. (2021), and Ergen & Pilanci (2021) from various viewpoints. On the other hand, the theory in the Axiotis & Sviridenko (2021) and the theory in the Woodworth et al. (2021) are not only based on convex optimization theory alone but also incorporate greedy, local search algorithms, and statistical knowledge, etc.

#### A.1.2 CONSTRAINED OPTIMIZATION-BASED THEORY

Different researchers approach the problem from diverse perspectives. Methods like Marsden et al. (2022) and the methods in Hu et al. (2020) focus on transforming constrained optimization problems. Some experts utilize geometric theory for this purpose, as demonstrated by Marsden et al. (2022) and Hu et al. (2020). In detail, the method in the Marsden et al. (2022) specifically addresses the minimization of Lipschitz convex functions on the unit ball, while the method in the Mhammedi (2022) explores the Frank-Wolfe algorithm in the context of constrained optimization, achieving certain levels of success.

#### A.1.3 GRADIENT DESCENT-BASED THEORY

Most of the existing methods are based on gradient descent theory and propose new algorithms such as Cheu et al. (2022),Garber & Kretzu (2022), and Fiez et al. (2022), and some endeavors focus on on theoretical innovation, such as Safran & Lee (2022), Vural et al. (2022) and Mladenovic et al. (2022). These articles delve into topics such as topology and geometry, exploring concepts like spherical indicator functions and flows in differential topology.

### A.2 OTHER THEORY

Researchers have approached this problem from different perspectives in fundamental mathematics, especially in topology, geometry, algebra, etc., and from the fields of statistics and dynamical system, etc., with some success.

Among the papers focusing on topology and geometry, the methods described in the Criscitiello & Boumal (2022) and Luo et al. (2022) explore the application of surface and curvature concepts. While the paper Douik & Hassibi (2022) and the paper Zhou et al. (2021) leverage the knowledge of Riemannian manifolds to address manifold optimization challenges.

From an algebraic perspective, papers such as Helala et al. (2022) and Xiong et al. (2022) propose algebraic frameworks for manifold optimization. In particular, a binary matrix optimization method is introduced in the Xiong et al. (2022), which provides experimental results to validate the theoretical correctness of the approach.

Additionally, statistical theory plays a significant role in manifold optimization. A range of methods based on statistical theory have been proposed, including works like Liu et al. (2022), Nie et al. (2022) and Xie et al. (2022).

### A.3 OPTIMIZATION METHODS

Researchers have been actively exploring strategies to achieve a balance between stability and accelerated in optimization methods. Notably, stochastic gradient descent (SGD) has emerged as a highly successful approach in numerous scientific and engineering domains.

### A.3.1 MACHINE LEARNING-BASED METHODS

The following articles are proposed within the framework of machine learning theory, with some focusing on sparse theory. Notably, the paper Zhang et al. (2021) and the paper Lyu et al. (2020) delve into sparse theory. Additionally, the paper Liu et al. (2019) explores the impact of learning rate on optimization, while the paper Exarchos et al. (2021) investigates the effects of changes in the objective function, yielding remarkable outcomes.

### A.3.2 DEEP LEARNING-BASED METHODS

Researchers have devoted substantial efforts to integrating optimization problems with deep learning, resulting in a range of compelling works. Noteworthy examples include Sahiner et al. (2022) and Wang et al. (2022).

## B PRELIMINARY:MANIFOLD THEORY

### B.1 CONCEPTIONS OF MANIFOLD

When it comes to manifold optimization, the concept of the manifold will be provided firstly. In mathematics, it is understand that a manifold in the Lee (2010) is a space that locally has the properties of a Euclidean space, and the properties of the whole space in terms of the local properties of the Euclidean space may be described. The canonical definition of a manifold is provided :

**Definition 3.** ManifoldLee (2010): It is determined a definition as the $n-$ dimensional topological manifold:

(i)The half space defined by $x_1 \geq 0$ is represented by $\mathbb{H}^n$, in the dimensional Euclidean space $\mathbb{R}^n$ ,

(ii)Hausdorff space $M$,

(iii)when each point $p$ has an open neighborhood $\mathscr{U}(p)$, with $\mathbb{R}^n$ or $\mathbb{H}^n$ homeomorphic.

Depending on the different classification methods, there are different manifolds, differential manifolds Boothby & Boothby (2003) are topological manifolds with differential structure, the manifolds for use in this paper are differential manifolds Boothby & Boothby (2003). Therefore the concepts of differential manifolds are assigned:

**Definition 4.** The topological space $(M, \mathscr{F})$ is defined as the $n-$ dimensional differentiable manifold Boothby & Boothby (2003)(also called smooth manifold): if $M$ has an open coverage $\mathscr{O}_a$, that is, $M = \mathscr{U}(\mathscr{O}_a)$.

A Riemannian manifold Lee (2006) is one of the differential manifolds Boothby & Boothby (2003) that defines a dot product in the tangent space Hu et al. (2020) at each point, and whose value changes smoothly with that point. Riemannian manifolds Lee (2006) permit us to determine numerous mathematical variables, such as the gradient of a function, the length and angle of an arc, as well as the area, volume, curvature, and scattering of a vector domain. The definition of a Riemannian manifold Lee (2006) is granted :

**Definition 5.** A Riemannian manifold Lee (2006) is a differential manifold of a Riemannian metric. Now the $M$ is set as an kind of $n-$ dimensional smooth manifold, then a tensor field $g$ is put on $M$, the tensor field $g$ is a second order covariant and smooth one, $(M, g)$ is an $n-$ dimensional Riemannian manifold, and $g$ is given a definition as a fundamental tensor or Riemannian metric of this Riemannian manifold and if it satisfies:

(i) $g(X, Y) = g(Y, X)(X, Y \in T_pM, p \in M)$, i.e, $g$ is symmetric.

(ii) $g(X, X) \geq 0(X \in T_pM, p \in M)$, i.e, $g$ is positive definite, and the equality sign holds only when $X = 0$.

Riemannian manifold Lee (2006) is a differential manifold that possesses additional geometric structures beyond the smoothness of its coordinate charts. This metric provides a geometric framework for various mathematical operations.

An isomorphism Sanjari et al. (2023) is interpreted as a state-project and there exists another state-project so that the composite of the two is a constant state-project. The main aim of investigating the

isomorphism Sanjari et al. (2023) is to extend the theory to different domains. If two structures are of isomorphism Sanjari et al. (2023), then the objects aboard them will share similar properties and operations, and propositions that are valid for one structure will be valid for the other. The definition of isomorphism Sanjari et al. (2023) is provided below:

**Definition 6.** An isomorphism Sanjari et al. (2023) between the group $(G_1, \star)$ and the group $(G_2, \star)$ is a one-to-one mapping compatible with the operator law. When a group is isomorphic to itself, it is referred to as a self-isomorphism Sanjari et al. (2023).

This is the basic concept of the manifold, if you would like to know more, please check it out in Hu et al. (2020).

### B.2 CONCEPTIONS OF REAL MANIFOLD

The manifold of Fixed Rank Matrices, denoted as $\mathbf{FR}(n, m)$ Vandereycken (2013), consist of $n \times m$ matrices with columns of unit norm. By careful consideration, it is endowed with a Riemannian manifold structure. Moreover, it is known to be a Riemannian submanifold embedded in the Euclidean space $\mathbb{R}^{n \times m}$, subject to the constraint that the rank of matrices in the manifold is $rank(X) = k$.

The manifold $V_{n,k}$ represents the set of orthonormal frames in an $n$-dimensional Euclidean space, where the frames consist of $k$ vectors. The Stiefel manifold Chen et al. (2023) is a real and compact analytic manifold, which can be associated with the classical compact groups $O(n)$ and $\mathrm{Sp}(n)$ as homogeneous spaces.

The Product Manifold Rovenski & Walczak (2023) is just a natural development of single manifold and product topology. So this concept is just briefly explained but ignore all the proofs with respect to its relevant arguments. The $\mathscr{M}_1$ manifold of the dimension is set as $d_1$, and $\mathscr{M}_2$ manifold of the dimension is $d_2$, respectively. The set $\mathscr{M}_1 \times \mathscr{M}_2$ is given a definition as set of pairs $(x_1, x_2)$, and it is known a conclusion that it can be described as $x_1 \in \mathscr{M}_1$ , $x_2 \in \mathscr{M}_2$, and if $(\mathscr{U}_1, \phi_1)$ are charts of the manifolds $\mathscr{M}_1$, $(\mathscr{U}_2, \phi_2)$ are charts of the manifolds $\mathscr{M}_2$, and the mapping $\phi_1 \times \phi_2 : (x_1, x_2) \to (\phi_1(x_1), \phi_2(x_2))$ is a chart for the set $\mathscr{M}_1 \times \mathscr{M}_2$. Analogously you could see that combining all charts could get an atlas. And then the maximal atlas. So the set $\mathscr{M}_1 \times \mathscr{M}_2$ becomes a manifold, which is defined as the product manifold, i.e, $\mathscr{M}_1$ and $\mathscr{M}_2$. The manifold topology of the product manifold bears comparison with the product topology.

The special orthogonal group Mahony et al. (2008) is a significant class of typical groups whose elements obtain determinant one. The elements of the orthogonal group $\mathbb{O}_n(\mathbf{K}, \mathbf{Q})$ all have determinant one or minus one, and all orthogonal transformations of which the determinant is one form a subgroup, a definition is just determined as special orthogonal group Mahony et al. (2008), made a contribution to $\mathbb{SO}_n(\mathbf{K}, \mathbf{Q})$. When the identity of $\mathbf{K} \neq 2$, $[\mathbb{O}_n(\mathbf{K}, \mathbf{Q}) : \mathbb{SO}_n(\mathbf{K}, \mathbf{Q})] = 2$. At this point $\mathbb{SO}_n(\mathbf{K}, \mathbf{Q})$ is also called the rotation group and is denoted as $\mathbb{O}_n(\mathbf{K}, \mathbf{Q})$. It is also the group consisting of the entire product of an even number of symmetries. The rotation group of a real orthogonal group $\mathbb{O}(n)$ is denoted as $\mathbb{SO}_n$ or $\mathbb{O}_n$. When identity of $\mathbf{K} = 2$, $\mathbb{SO}_n(\mathbf{K}, \mathbf{Q}) = \mathbb{O}_n(\mathbf{K}, \mathbf{Q})$. Let $\mathbf{Q}$ have no loss number, i.e. $f(x, y) = \mathbf{Q}(x + y) - \mathbf{Q}(x) - \mathbf{Q}(y)$, which is not a degenerate intersection type on space $\mathbb{V}$ given a definition by $\mathbf{Q}$.

The manifold of symmetric positive definite matrices Jayasumana et al. (2013) is of the bivariant geometry. The function $M$ can be represented as $sym positive definite factory(n)$. A point $X$ on the manifold is expressed as a symmetric positive definite matrix $X_{n \times n}$. The Riemannian metric has the bi-invariant metric, whose tangent vectors are symmetric matrices.

The Euclidean manifold Phogat & Chang (2022) may be determined a definition on an Euclidean space. An Euclidean space is a finite real vector space $\mathbb{R}^n$ whose inner product is $(x, y)$, $x, y \in \mathbb{R}^n$, it is in an appropriate(Cartesian) coordinate system $x = (x_1, \dots, x_n)$ and $y = (y_1, \dots, y_n)$ is provided by the formula:

$$(x, y) = \sum_{i=1}^{n} x_i y_i. \tag{11}$$

For more details, please consult the paper Hu et al. (2020).

## C   METHODOLOGY: RESET METHOD

### C.1   PROPOSED METHODS

#### C.1.1   THEOREMS OF RESET METHOD ON THE REAL MANIFOLDS

As for convergence, known from (Hu et al. (2020)) that the step size of manifold optimization is convergent, so whether or not the method converges after improving it, and whether there is a logically clear proof, theorem 4.1 returns to this question.

**Theorem C.1.** Let $x_i$ be the sequence generated by Reset method on the real manifolds using non-monotonic sequences, it is assumed that $f$ is continuously differential on the real manifold $\mathscr{M}$ and Euclidean space $\mathscr{E}$. In this context, every accumulation point $x^*$ of the sequence $x_i$ is considered a stationary point of the optimization problem, i.e., it holds $\mathrm{gradf}(\mathrm{x}^*) = 0$.

*Proof.* By using
$$\langle \mathrm{gradf}(\mathrm{x_i}), -\mathrm{gradf}(\mathrm{x_i}) \rangle_{x_i} = -\|\mathrm{gradf}(\mathrm{x_i})\|^2_{x_i} < 0,$$
then applying Lemma 1.1 in the Zhang & Hager (2004), it is possible to:
$$f(x_i) \le C_i, \forall i \in \mathbb{N}, x_i \in \{x \in \mathscr{M} : f(x) \le f(x_0)\}.$$

Due to
$$\lim_{t \to 0} \frac{(f \circ R(x_i)(-t \times \mathrm{gardf}(\mathrm{x_i})) - f(x_i)}{t} - \alpha \langle \mathrm{gradf}(\mathrm{x_i}), -\mathrm{gradf}(\mathrm{x_i}) \rangle_{x_i}$$
$$= \frac{\partial f(R(x_i)(0))^T}{\partial x} dR(x_i)(0)(-\mathrm{gradf}(\mathrm{x_i})) + \alpha\|\mathrm{gradf}(\mathrm{x_i})\|^2_{x_i}$$
$$= -(1-\alpha)\|\mathrm{gradf}(\mathrm{x_i})\|^2_{x_i} < 0,$$

there always exists a positive step size satisfying the Armijo conditions (eq. (6) and eq. (7) ), Let $x^* \in \mathscr{M}$ be an arbitrary accumulation point of $x_i$ and let $x_{ii}$ be a corresponding subsequence converge to $x^*$. By the definition of $C_{i+1}$ , it is possible to:
$$C_{i+1} = \frac{\zeta R_i C_i + f(x_{i+1})}{R_{i+1}} < \frac{(\zeta R_i + 1)C_i}{R_{i+1}} = C_i.$$

$C_i$ is monotonically decreasing while converging to some limit $\overline{C} \in \mathbb{R} \bigcup -\infty$. Using $\forall i \to \infty, f(x_i) \to f(x^*)$, it can be deduced that $\overline{C} \in \mathbb{R}$. Therefore, it is possible to:
$$\sum_{i=0}^{\infty} \frac{\alpha(-\alpha_i)\|\mathrm{gradf}(\mathrm{x_i})\|^2_{x_i}}{R_{i+1}} \le \sum_{i=0}^{\infty} C_i - C_{i+1} \le C_0 - \overline{C} < \infty.$$

Since
$$R_{i+1} = 1 + \zeta R_i = 1 + \zeta + \zeta^2 R_{i-1} = \sum_{i=0}^{i} \zeta^i < (1-\zeta)^{-1},$$

this means $\{-\alpha_i\|\mathrm{gradf}(\mathrm{x_i})\|^2_{x_i}\} \to 0$. Let us assume $\|\mathrm{gradf}(\mathrm{x}^*)\| \ne 0$, it is possible that $t_{ii} \to 0$, by the construction of Reset method on the real manifolds, the step size does not satisfy eq. (7), i.e., when $i \to +\infty$, it is possible to
$$\alpha \epsilon^{-1} \alpha_i \|\mathrm{gradf}(\mathrm{x_i})\|^2_{x_i} < f(R(x_i)(\epsilon^{-1}\alpha_i\mathrm{gradf}(\mathrm{x_i})) - C_i \le f(R(x_i)(\epsilon^{-1}\alpha_i\mathrm{gradf}(\mathrm{x_i})) - f(x_i).$$

Due to the sequence $\mathrm{gradf}(\mathrm{x_i})_i$ is bounded, the rest of the proof is the same as the proof of Theorem in Tunçel (2009). □

It is possible to already known from theorem 4.1 that our proposed algorithm converges, and the Reset method improves convergence since it can produce damped harmonic motion Yadav et al. (2018) that sinks into the saddle point. From our knowledge of the dynamical system Yadav et al. (2018), it is understand that the damping produced by the Reset method causes the orbit to converge to the saddle point and the error decays at an exponential rate.

**Theorem C.2.** Let $x_i$ be the sequence generated by Reset method on the real manifolds using a non-monotonic sequence. It is assumed that the objective function $f$ is continuously differentiable on the real manifold $\mathscr{M}$. Every accumulation point $x^*$ of the sequence $x_i$ is considered a stationary point of the optimization problem. Let $x^\star$, $x^\heartsuit$ and $x^\diamond$ be a point obtained after backward propagation of the gradient using other methods (SGD, Adam Kingma & Ba (2014) and AdamW Loshchilov & Hutter (2017)) respectively. Furthermore, it is assumed that there exists a stochastic gradient which satisfies $\mathbb{E}[\|\mathrm{gradf}(x_i^*)\|^2] \leq \epsilon_0^2$, $\mathbb{E}[\|\mathrm{gradf}(x_i^\star)\|^2] \leq \epsilon_1^2$, $\mathbb{E}[\|\mathrm{gradf}(x_i^\heartsuit)\|^2] \leq \epsilon_2^2$, $\mathbb{E}[\|\mathrm{gradf}(x_i^\diamond)\|^2] \leq \epsilon_3^2$, the error bound is that:

$$\mathbb{E}[\mathrm{gradf}(x_{i+1}^\star)] - \mathbb{E}[\mathrm{gradf}(x_i)] \leq \frac{1}{2}\left(\frac{4i^2}{(i+1)^2}\epsilon_1^2 + \epsilon_0^2\right),$$

$$\mathbb{E}[\mathrm{gradf}(x_{i+1}^\heartsuit)] - \mathbb{E}[\mathrm{gradf}(x_i)] \leq \frac{1}{2}\left(\frac{4i^2}{(i+1)^2}\epsilon_2^2 + \epsilon_0^2\right), \tag{12}$$

$$\mathbb{E}[\mathrm{gradf}(x_{i+1}^\diamond)] - \mathbb{E}[\mathrm{gradf}(x_i)] \leq \frac{1}{2}\left(\frac{4i^2}{(i+1)^2}\epsilon_3^2 + \epsilon_0^2\right).$$

*Proof.* Part I: Due to

$$\mathbb{E}[\mathrm{gradf}(x_{i+1}^\star)] = \frac{1}{i+1}\left(i \times \mathbb{E}[\mathrm{gradf}(x_i^\star)] + \mathrm{gradf}(x_{i+1}^\star)\right), \tag{13}$$

and the loss function is monotonically decreasing, then $\mathrm{gradf}(x_{i+1}^\star) \leq \mathrm{gradf}(x_i^\star)$, and $\mathrm{gradf}(x_i^\star) \leq \mathbb{E}[\mathrm{gradf}(x_i^\star)]$, due to

$$\begin{aligned} &\mathbb{E}[\mathrm{gradf}(x_{i+1}^\star)] - \mathbb{E}[\mathrm{gradf}(x_i)] \\ =&\frac{1}{i+1}\left(\mathbb{E}[\mathrm{gradf}(x_i^\star)] + \mathrm{gradf}(x_{i+1}^\star)\right) - \mathbb{E}[\mathrm{gradf}(x_i)] \\ \leq&\frac{i}{i+1}\left(\mathbb{E}[\mathrm{gradf}(x_i^\star)] + \mathrm{gradf}(x_i^\star)\right) - \mathbb{E}[\mathrm{gradf}(x_i)] \\ \leq&\frac{i}{i+1}\left(2 \times \mathbb{E}[\mathrm{gradf}(x_i^\star)]\right) - \mathbb{E}[\mathrm{gradf}(x_i)] \\ \leq&\mathbb{E}[\langle \frac{2i}{i+1}\mathrm{gradf}(x_i^\star), \mathrm{gradf}(x_i)\rangle] \\ \leq&\mathbb{E}[\|\frac{2i}{i+1}\mathrm{gradf}(x_i^\star)\|\|\mathrm{gradf}(x_i)\|] \\ \leq&\frac{1}{2}\left(\mathbb{E}[\|\frac{2i}{i+1}\mathrm{gradf}(x_i^\star)\|^2] + \mathbb{E}[\|\mathrm{gradf}(x_i)\|^2]\right) \\ \leq&\frac{1}{2}\left(\frac{4i^2}{(i+1)^2}\mathbb{E}[\|\mathrm{gradf}(x_i^\star)\|^2] + \mathbb{E}[\|\mathrm{gradf}(x_i)\|^2]\right) \\ \leq&\frac{1}{2}\left(\frac{4i^2}{(i+1)^2}\epsilon_1^2 + \epsilon_0^2\right). \end{aligned} \tag{14}$$

Part II: It is known that

$$\mathbb{E}[\mathrm{gradf}(x_{i+1}^\heartsuit)] = \frac{1}{i+1}\left(i \times \mathbb{E}[\mathrm{gradf}(x_i^\heartsuit)] + \mathrm{gradf}(x_{i+1}^\heartsuit)\right), \tag{15}$$

and the loss function is monotonically decreasing, then $\mathrm{gradf}(x_{i+1}^\heartsuit) \leq \mathrm{gradf}(x_i^\heartsuit)$, and $\mathrm{gradf}(x_i^\heartsuit) \leq \mathbb{E}[\mathrm{gradf}(x_i^\heartsuit)]$, inspired by the eq. (14), it is possible to:

$$\begin{aligned} &\mathbb{E}[\mathrm{gradf}(x_{i+1}^\heartsuit)] - \mathbb{E}[\mathrm{gradf}(x_i)] \\ \leq&\frac{1}{2}\left(\frac{4i^2}{(i+1)^2}\mathbb{E}[\|\mathrm{gradf}(x_i^\heartsuit)\|^2] + \mathbb{E}[\|\mathrm{gradf}(x_i)\|^2]\right) \\ \leq&\frac{1}{2}\left(\frac{4i^2}{(i+1)^2}\epsilon_2^2 + \epsilon_0^2\right). \end{aligned} \tag{16}$$

Part III: It is known that

$$\mathbb{E}[\mathrm{gradf}(x_{i+1}^{\diamond})] = \frac{1}{i+1}(i \times \mathbb{E}[\mathrm{gradf}(x_i^{\diamond})] + \mathrm{gradf}(x_{i+1}^{\diamond})), \qquad (17)$$

and the loss function is monotonically decreasing, then $\mathrm{gradf}(x_{i+1}^{\diamond}) \leq \mathrm{gradf}(x_i^{\diamond})$, and $\mathrm{gradf}(x_i^{\diamond}) \leq \mathbb{E}[\mathrm{gradf}(x_i^{\diamond})]$, inspired by the eq. (14), it is possible to:

$$\begin{aligned}
&\mathbb{E}[\mathrm{gradf}(x_{i+1}^{\diamond})] - \mathbb{E}[\mathrm{gradf}(x_i)] \\
\leq &\frac{1}{2}(\frac{4i^2}{(i+1)^2}\mathbb{E}[\|\mathrm{gradf}(x_i^{\diamond})\|^2] + \mathbb{E}[\|\mathrm{gradf}(x_i)\|^2]) \\
\leq &\frac{1}{2}(\frac{4i^2}{(i+1)^2}\epsilon_3^2 + \epsilon_0^2).
\end{aligned} \qquad (18)$$

$\square$

## D  DEEP LEARNING EXPERIMENTS

### D.1  DATASETS

The CIFAR-100 Krizhevsky et al. (2009) dataset has 100 classes containing 600 images each, and each of size is $32 \times 32$. There are 500 training images and 100 testing images per class. The 100 classes in the CIFAR-100 Krizhevsky et al. (2009) dataset are grouped into 20 superclasses. Each image comes with a "fine" label (the class to which it belongs) and a "coarse" label (the superclass to which it belongs).

In the STL-10 dataset, there are 500 training images and 800 test images for each category. The unlabelled dataset comprises 100,000 unlabelled images, including animals and vehicles in categories other than the 10 categories. All images are sourced from the ImageNet.

The SVHN (Street View House Numbers) dataset is a real-world dataset for numerical recognition of street house numbers, which contains two formats: full numbers and cropped digit. The Cropped Digit format is a color image cropped to $32 \times 32$, the training set contains 73257 images, the test set contains 26,032 images, and there is an extra training set containing 531131 images. an extra training set containing 531131 images. SVHN dataset is door number digits extracted from Google Street View images for developing machine learning and object recognition algorithms.

The CIFAR-10 Krizhevsky et al. (2009) dataset contains 10 RGB colour image categories, each of size $32 \times 32$, with 6,000 images in each category, and 5,000 training images and 10,000 test images in the dataset.

The Market-1501 Zheng et al. (2015) dataset is collected from Tsinghua University campus, including 1501 persons, 32668 detected person rectangular frames. This dataset has 12,936 training images and 19,732 test images. The catalogue structure of the Market-1501 Zheng et al. (2015) dataset consists of a training set, a test set, and a query set, in which the training set contains 751 images of pedestrians, the test set contains 750 images of persons, and the query set contains 3,368 manually drawn pedestrian detection rectangles, the test set contains 750 person images, and the query set contains 3368 manually drawn rectangular box images for person detection. This dataset is collected to evaluate the performance of the person re-identification algorithm.

The DukeMTMC-reID Zheng et al. (2017) dataset is a large-scale person re-identification image dataset, which is collected by Duke University specifically for person re-identification (ReID) research. The dataset consists of 16,522 training images, 2,228 query images and 17,661 gallery images involving 702 persons. In addition, the DukeMTMC-reID Zheng et al. (2017) dataset provides manually labelled bounding boxes, which are useful for training and testing person detection algorithms. Its wide application and recognition proves its significant value and influence in the field of person re-ID.

Table 10: Image Generation of the Deep Convolutional Generative Adversarial Network (DCGAN) Yadav et al. (2017) on STL-10 dataset

| Methods | Average Precision(AP) |
| --- | --- |
| Adam Kingma & Ba (2014) | 98.59 |
| Adam Kingma & Ba (2014) + e(Ours) | 99.56 |
| Adam Kingma & Ba (2014) + fr(Ours) | 99.58 |
| Adam Kingma & Ba (2014) + g(Ours) | 99.59 |
| Adam Kingma & Ba (2014) + o(Ours) | 99.61 |
| Adam Kingma & Ba (2014) + p(Ours) | 99.59 |
| Adam Kingma & Ba (2014) + psd(Ours) | 99.62 |
| Adam Kingma & Ba (2014) + s (Ours) | 99.58 |
| Adam Kingma & Ba (2014) + sog (Ours) | 99.62 |
| Adam Kingma & Ba (2014) + sp (Ours) | 99.65 |
| Adam Kingma & Ba (2014) + spv(Ours) | **99.71** |
| AdamW Loshchilov & Hutter (2017) | 98.52 |
| AdamW Loshchilov & Hutter (2017) + e (Ours) | 99.56 |
| AdamW Loshchilov & Hutter (2017) + fr (Ours) | 99.61 |
| AdamW Loshchilov & Hutter (2017) + g (Ours) | 99.75 |
| AdamW Loshchilov & Hutter (2017) + o (Ours) | 99.65 |
| AdamW Loshchilov & Hutter (2017) + p (Ours) | 99.75 |
| AdamW Loshchilov & Hutter (2017) + psd (Ours) | 99.72 |
| AdamW Loshchilov & Hutter (2017) + s (Ours) | 99.68 |
| AdamW Loshchilov & Hutter (2017) + sog (Ours) | 99.74 |
| AdamW Loshchilov & Hutter (2017) + sp(Ours) | 99.76 |
| AdamW Loshchilov & Hutter (2017) + spv(Ours) | **99.78** |
| SGD | 98.27 |
| SGD + e (Ours) | 99.72 |
| SGD + fr (Ours) | 99.66 |
| SGD + g (Ours) | 99.63 |
| SGD + o (Ours) | 99.69 |
| SGD + p (Ours) | 99.63 |
| SGD + psd (Ours) | 99.68 |
| SGD + s (Ours) | 99.63 |
| SGD + sog (Ours) | 99.68 |
| SGD + sp (Ours) | 99.62 |
| SGD + spv (Ours) | **99.69** |

## D.2 DEEP LEARNING EXPERIMENTS

### D.2.1 DEEP CONVOLUTIONAL GENERATIVE ADVERSARIAL NETWORK (DCGAN) FOR IMAGE GENERATION

The STL-10 dataset and the SVHN (Street View House Numbers) dataset are selected as a measure of domain adaptation performance. Our methods are evaluated on the Deep Convolutional Generative Adversarial Network (DCGAN) Yadav et al. (2017) backbone with an initial learning rate of 0.0002, and the model is trained with 5,000 iterations on 4 V100 GPUs at a scale of 32 batches and the learning rate warmup He et al. (2019) is employed.

Through experiments, the results confirm our intuition and validated the effectiveness and stability of our methods. The average precision performance is summarized in table 10 and table 11, from the table, it is known that our methods are more stable and shows a higher average precision than the competitors.

Table 11: Image Generation of the Deep Convolutional Generative Adversarial Network (DCGAN) Yadav et al. (2017) on SVHN dataset

| Methods | Average Precision(AP) |
| --- | --- |
| Adam Kingma & Ba (2014) | 98.62 |
| Adam Kingma & Ba (2014) + e(Ours) | 99.82 |
| Adam Kingma & Ba (2014) + fr(Ours) | 99.31 |
| Adam Kingma & Ba (2014) + g(Ours) | 99.42 |
| Adam Kingma & Ba (2014) + o(Ours) | 99.46 |
| Adam Kingma & Ba (2014) + p(Ours) | 99.51 |
| Adam Kingma & Ba (2014) + psd(Ours) | 99.58 |
| Adam Kingma & Ba (2014) + s (Ours) | 99.61 |
| Adam Kingma & Ba (2014) + sog (Ours) | 99.65 |
| Adam Kingma & Ba (2014) + sp (Ours) | 99.68 |
| Adam Kingma & Ba (2014) + spv(Ours) | **99.72** |
| AdamW Loshchilov & Hutter (2017) | 98.47 |
| AdamW Loshchilov & Hutter (2017) + e (Ours) | 99.56 |
| AdamW Loshchilov & Hutter (2017) + fr (Ours) | 99.61 |
| AdamW Loshchilov & Hutter (2017) + g (Ours) | 99.75 |
| AdamW Loshchilov & Hutter (2017) + o (Ours) | 99.86 |
| AdamW Loshchilov & Hutter (2017) + p (Ours) | 99.95 |
| AdamW Loshchilov & Hutter (2017) + psd (Ours) | 99.25 |
| AdamW Loshchilov & Hutter (2017) + s (Ours) | 99.38 |
| AdamW Loshchilov & Hutter (2017) + sog (Ours) | 99.44 |
| AdamW Loshchilov & Hutter (2017) + sp(Ours) | 99.56 |
| AdamW Loshchilov & Hutter (2017) + spv(Ours) | **99.77** |
| SGD | 98.35 |
| SGD + e (Ours) | 99.62 |
| SGD + fr (Ours) | 99.66 |
| SGD + g (Ours) | 99.63 |
| SGD + o (Ours) | 99.69 |
| SGD + p (Ours) | 99.63 |
| SGD + psd (Ours) | 99.66 |
| SGD + s (Ours) | 99.62 |
| SGD + sog (Ours) | 99.68 |
| SGD + sp (Ours) | 99.62 |
| SGD + spv (Ours) | **99.69** |