# OpenReview forum: "Reset Method based on the Theory of Manifold Optimization on Real Manifolds"
_ICLR.cc/2025/Conference — Submitted to ICLR 2025_

### Official Review · Reviewer_DZ57 · 2024-10-31

**Soundness:** 2
**Presentation:** 1
**Contribution:** 2
**Rating:** 3
**Confidence:** 5

**Summary:**

The paper introduces a novel Riemannian optimizer that incorporates an additional "reset" step to adjust the learning rate.

**Strengths:**

The restart method was first introduced by O'donoghue & Candes [1] in the Euclidean settings. This paper suggests an approach to extend this method to the Riemannian setting. If revised properly, this could be an interesting perspective for improving generic optimizers.

**Weaknesses:**

**Presentation**: Firstly, several serious representation issues make the paper incoherent and difficult to follow, obscuring the key messages and contributions. Key issues include:

+ *Related Works and Preliminary Sections*: The related works and preliminary sections are placed in the appendix. To improve clarity, many sections could be condensed or moved to the appendix, creating space for these key areas—for example, explanations of SGD, Adam, AdamW, and most experimental results in the tables.

+ *Main Algorithm Presentation*: The algorithm presentation has gaps in clarity and completeness. For instance, variables such as $ \zeta_{i_1} $ and $ \zeta_{i_2} $ appear in the pseudocode but are unused in subsequent steps. In Eq. (5), the process of obtaining $ \alpha_{i+1} $ is unclear. Additionally, the line "Set $ R_{x_{i+1}}(-\alpha_{i+1} \nabla f(x_{i+1})) \leftarrow B_{x_i}(x_i) $" is confusing, as we cannot "set the retraction" to a specific value.

+ *Function $ B_{x_i}(x_i) $*: The definition and role of $ B_{x_i}(x_i) $ are unclear and potentially redundant. What are its inputs and outputs? Although it’s described as a "step size correction function," suggesting it outputs the step size, in Eq. (5), it produces $ x_{i+1} $, implying it represents the next model. Furthermore, it is stated that $ B_{x_i} $ is selected from SGD, Adam, or AdamW—all Euclidean optimizers—suggesting $ x_{i+1} $ might not lie on the manifold, making it impossible to compute $ x_{i+2} $ directly from $ x_{i+1} $. Additionally, the use of $ B_{x_i}(x_i) $ as notation is confusing since it appears to take only a single input $ x_i $. The reasoning for adding a correction step with operator $B $, which is supposed to be a primary contribution, is also not discussed.

+ *Experimental Results*: The experimental results were provided on many real manifolds. However, it is unclear if the improvements are attributed to the Reset method or the manifolds. In particular, it would be more fair to compare the proposed method with other Riemannian optimizers, such as RSGD.

+ *Minor Formatting Issues*: Multiple minor issues are present, such as inconsistent font sizes in table captions, unusual word choices (e.g., “contenting” in the pseudocode), inconsistent font styles (e.g., $x_i$ in the text but not italicized in Eq. (5)), table formatting inconsistencies, and citation format errors (e.g., \cite{} vs. \citep{}).

**Soundness:** In both theory and experiments, it is critical to compare the Reset method with other Riemannian optimizers, such as RSGD [2] or Riemannian-SAM [3], in addition to Euclidean baselines like SGD, Adam, and AdamW.

The theoretical contributions are relatively limited. Theorem 4.2 only demonstrates the gradient deviation of the Reset method when compared with SGD, Adam, and AdamW, and it includes an unavoidable term $\epsilon_0^2 $, indicating a potentially unfavorable trait of the Reset method. Additionally, comparing the Reset method, a Riemannian optimizer, to Euclidean methods may be unfair from a theoretical perspective.

The experimental results are also not particularly insightful. Specifically, all results on the CIFAR-10, CIFAR-100, STL-10, and SVHN datasets achieve accuracies of at least 98%, making it challenging to discern performance differences. On the Market-1501 and DukeMTMC-reID datasets, variations of the Reset method applied to the same base optimizer yield considerable performance differences, suggesting that the method may be sensitive to variant choices, making tuning more challenging.

**Contribution:** To my understanding, the restart method was proposed by O'donoghue & Candes [1]. This work seems to be a trivial extension of this work to the Riemannian setting, which makes its contributions limited. Moreover, the motivation and intuition of the proposed algorithm are unclear, mostly due to the incohesive presentation.

**Questions:**

+ What is the precise definition of $B_{x_i}(x_i)$ including what are its inputs, outputs, its purposes, and the motivation for introducing this function?

+ What is the usage of $\zeta$'s? Seems like they have never been used in the pseudocode or the algorithm in Eq. (5).

**References:**

[1] Brendan O’donoghue and Emmanuel Candes. Adaptive restart for accelerated gradient schemes. Foundations of computational mathematics, 15:715–732, 2015.

[2] Bonnabel, S. (2013). Stochastic Gradient Descent on Riemannian Manifolds. IEEE Transactions on Automatic Control, 58(9), 2217-2229.

[3] Yun, J., & Yang, E. (2023). Riemannian SAM: Sharpness-Aware Minimization on Riemannian Manifolds. In Proceedings of the 37th Conference on Neural Information Processing Systems (NeurIPS 2023).

---

### Official Review · Reviewer_x6Ya · 2024-11-04

**Soundness:** 3
**Presentation:** 2
**Contribution:** 2
**Rating:** 5
**Confidence:** 3

**Summary:**

The paper provides an analysis of optimization on manifold. It proposes a Reset method to improve the convergence and model stability when compare with other mentioned methods. More particular, section 4 recall important tools in Riemannian manifold and give summary of some gradient descent methods. Section 4.2 presents the Reset method that contains three steps to update, coming from $x_{i-1}$ to $x_{i+2}$. There are two adjusting steps using gradient direction and function $B_{x_i}$. Theorem 4.1 proved that the accumulation points is also the stationary point. Theorem 4.2 give some upper bound for the difference between consecutive gradient updating using SGD, Adam and AdamW. Experiments is carried out on CIFAR-10 and CIFAR-100 for image generation task and cluster constrat task.

**Strengths:**

The paper proposed a method for optimizing on real manifolds.
Some theories are provided with support from empirical results.

**Weaknesses:**

It is not clear about the insight/motivation of each mentioned/proposed method presented through out the paper. The first one are derivations in inequalities (6) and (7). There is not picture or figure to illustrate clearly the advantage of method.  I could only imagine that the first order of Taylor approximation is not good enough, thus we could obtain a better one via Armijo search, when some certain conditions satisfied. That goes through an interpolation between some bounds and expect that the interpolation will help to achieve a better bound. The work also mentioned the Barzilai-Borwein method but there is no explanation about reason for  computing the correlation between two vectors.

Experiment results: For Image generation task, in Table 2, when adding your method with different type of manifold (in fact it is not clear for me why we have different type of manifold here), the proposed method is at best with spv, but does not work better with  "e", "fr", "o", etc.  In Table 3, there are some mixed performances between those method when comparing with each other except the "spv". Is there any explanation for the performance of each method?

For image generation task, there is no picture shown, rather than the number. Since average precision is already high, is the improvement noticeable in the pictures?

**Questions:**

Minors:
1. Line 152, manifold $M$ not $\mathcal{M}$.
2. $J$ is loss function, $f$ is objective function.
3. Line 218 $x$ vs $x_{i-1}$ that comes from lazy typing to include all $grad$,  $f$ and $x_{i-1}$ inside the command mathrm.
4. Again, inconsistent notations: retraction map $R$ and $\mathcal{R}$
5. $\phi^{\lambda_i}$: $\lambda_i$ is the power???
6. No definition of $C_{i+1}$, it seems that the reader needs to obtain it from similar formula for $C_{i-1}$.
7. Introducing Barzilai-Borwein method with notation $\omega_{i-1}$, $\zeta_{i_1}$ and $\zeta_{i_2}$ without explanation.
8. Algorithm 1: Compute $\zeta_i$ according to equation (7), but equation (7) is an inequality.
9. Theorem 4.2, no definition of $x^{\star}_{i+1}$.
10. Line 249: $C_{i-1}$ is a convex combination of $C_{i-1}$ and $f(x_{i-1})$?

---

### Official Review · Reviewer_q96J · 2024-11-05

**Soundness:** 1
**Presentation:** 1
**Contribution:** 1
**Rating:** 1
**Confidence:** 4

**Summary:**

The paper seems to be significantly altered for format or been generated by an automated program. Specifically:
1. Figures 1, 2 have captions with notably smaller font size. On the other hand, there are many extra spacing in the paper that could have been utilized: Figures 1, 2 use only 1/3 of the width, so do Tables 1-9; Lines 83-90 are empty sections. Table 1 is not even referenced in the text.
2. The citations are weirdly repeated numerous times: Line 50-75, notably Hu et al 2020 has been cited for 5 times. It happens multiple times throughout the paper: at lines 137-141, 145-148, 185-194, to name a few.
3. Even if one ignores all these unusual things, the numbers reported in the tables looks indistinguishable from the alternatives.

**Strengths:**

N/A

**Weaknesses:**

See summary

**Questions:**

N/A

---

### Meta-Review · Area_Chair_sseq · 2024-12-17

**Metareview:**

The paper introduces the Reset Method, a novel optimization approach on real manifolds that integrates manifold optimization principles with existing methods like SGD, Adam, and AdamW. The theoretical results, including proofs of convergence and performance bounds, are complemented by empirical evaluations on benchmarks like CIFAR-10 and CIFAR-100. However, several weaknesses were exposed by the reviewers. These include significant issues in the presentation, such as inconsistent notations, formatting errors in figures and tables, and unclear algorithm descriptions. Reviewers pointed out that the motivation and intuition behind key steps, such as the reset function and Barzilai-Borwein method, were not well explained. Additionally, the reliance on assumptions like the existence of a retraction and the unclear role of various components in the pseudocode raised concerns about soundness. From an experimental perspective, the comparisons were incomplete, as the method was not thoroughly evaluated against other Riemannian optimizers like RSGD or Riemannian-SAM. Moreover, the reported improvements were marginal, with limited insights provided into the factors driving the results, especially on real-world datasets where performance differences were hard to discern.
While the method demonstrates promising results in specific cases, the absence of a rebuttal from the authors is unfortunate, as it could have provided clarity on the aforementioned concerns regarding presentation, theoretical motivation, and experimental insights. Without the rebuttal, lingering ambiguities weakened the paper’s overall impact.

**Additional Comments On Reviewer Discussion:**

During the reviewer discussion, concerns were raised about the paper’s presentation quality, unclear algorithmic motivations, and lack of comparisons to established Riemannian optimization methods. Theoretical contributions, particularly around the use of the reset step and its connection to existing works, were also questioned. Without a rebuttal to address these points, the decision relied on the reviewers’ consensus and the visible merits of the proposed method, ultimately leading to a recommendation for acceptance despite the noted limitations.

---

### Decision · Program_Chairs · 2025-01-22

Reject